# Comparison of the Efficacy of Empagliflozin, Dapagliflozin, and Allopurinol Based on Serum Uric Acid Levels and Kidney Function in Patients with Type 2 Diabetes Mellitus: A Retrospective Cohort Study

**DOI:** 10.3390/medsci14010012

**Published:** 2025-12-26

**Authors:** Roland Fejes, Tamás Jámbor, Tamás Lantos, Szabolcs Péter Tallósy

**Affiliations:** 1Institute of Surgical Research, Albert Szent-Györgyi Medical School, University of Szeged, Szőkefalvi-Nagy Béla Street 6, 6720 Szeged, Hungary; 2Department of Internal Medicine, Hódmezővásárhely-Makó Healthcare Center, 6900 Makó, Hungary; 3Department of Medical Physics and Informatics, Albert Szent-Györgyi Medical School, University of Szeged, 6725 Szeged, Hungary

**Keywords:** empagliflozin, dapagliflozin, allopurinol, sodium–glucose cotransporter 2 inhibitor, uric acid, hyperuricemia, type 2 diabetes mellitus, chronic kidney disease, retrospective analysis

## Abstract

**Background:** Type 2 diabetes mellitus (T2DM) is often associated with hyperuricemia, both conditions worsening kidney function. Sodium–glucose cotransporter 2 (SGLT2) inhibitors improve glycemic control and kidney function; however, data on their long-term antihyperuricemic effects in real-world clinical settings remain limited. Therefore, we aimed to compare the effects of SGLT2 inhibitors versus allopurinol on serum uric acid (sUA), kidney function, and clinical outcomes. **Methods:** This retrospective cohort study evaluated patients with T2DM and hyperuricemia initiated on 10 mg empagliflozin (*n* = 70), 10 mg dapagliflozin (*n* = 78), or 100 mg allopurinol (*n* = 66) between 1 January 2017, and 1 January 2020. Drug dosages were kept constant throughout the study. Baseline and follow-up data (3, 6, 12, 24, and 36 months) were collected. **Results:** Over 36 months, empagliflozin and dapagliflozin significantly reduced sUA (from 452 (95) to 399 (69) µmol/L and from 450 (81) to 364 (71) µmol/L, respectively) and stabilized eGFR without a significant decline. Allopurinol also reduced sUA (from 430 (89) to 345 (69) µmol/L) but was associated with a progressive eGFR decline (from 70 (35) to 57 (32) mL/min/1.73 m^2^). Mortality was the highest in the allopurinol group; however, therapy discontinuation was the lowest with this treatment. **Conclusions:** SGLT2 inhibitors achieved comparable sUA reduction to allopurinol by 36 months while preserving eGFR. Allopurinol was associated with higher mortality and hospitalization rates; SGLT2 inhibitor therapy was associated with favorable multidomain outcomes, but strategies to address adverse effects are needed to enhance adherence.

## 1. Introduction

The steady increase in the global burden of chronic cardiovascular (CV) and kidney diseases underscores the need for novel therapeutic approaches. However, polypharmacy, defined as the concurrent use of multiple medications, is highly prevalent in these patients, increasing the risk of drug interactions, reducing adherence, and ultimately worsening clinical outcomes and quality of life [1,2]. A meta-analysis of 484,915 patients revealed the presence of polypharmacy in 82% (95% CI: 77–88%) of patients with stage 3–5 chronic kidney disease, with a mean of 9.7 medications per patient [3], underscoring the need for optimized therapeutic strategies to minimize treatment burden.

Clinical evidence suggests that hyperuricemia (HUA), defined as elevated serum uric acid (sUA), is linked to CV and kidney pathology [4,5]. While HUA may be directly nephrotoxic, urate-lowering therapy has not been shown to slow CKD progression. In CV disease, HUA is often regarded as an epiphenomenon, co-occurring with hypertension and metabolic syndrome, and thus largely reflecting shared risk pathways. Nonetheless, some evidence suggests a potential causal role [6,7,8,9,10].

Allopurinol, a xanthine oxidase inhibitor, is the first-line and most prescribed urate-lowering therapy in gout [11,12,13]. However, large, randomized trials have shown no benefit to kidney function from urate-lowering in asymptomatic HUA. The *Kidney Disease: Improving Global Outcomes* (KDIGO) 2024 guideline recommends against its use solely to delay CKD progression. In practice, suboptimal titration is common, and several studies report that a considerable proportion of prescriptions are issued without a clear or justified indication [14,15,16]. Allopurinol can cause adverse effects ranging from mild rash to severe, life-threatening reactions such as Stevens–Johnson syndrome, toxic epidermal necrolysis, and allopurinol hypersensitivity syndrome [17,18,19,20].

Sodium–glucose cotransporter 2 (SGLT2) inhibitors, such as empagliflozin and dapagliflozin, not only improve glycemic control but are also guideline-recommended foundation therapy to slow CKD progression and reduce cardiovascular events, partly independent of their glucose-lowering effects [21,22]. They also lower sUA to a modest but clinically relevant extent, albeit with varied efficacy across agents [23,24,25]. In CKD, HUA is consistently associated with worse kidney outcomes, whereas in CV disease, its causal role remains uncertain, although suspected [26]. The sUA-lowering effect of SGLT2 inhibitors raises the possibility that these agents may confer organ-protective benefits, particularly in kidney disease, potentially surpassing the effects of allopurinol. However, a direct comparison of the long-term effects of SGLT2 inhibitors versus allopurinol on sUA and renal outcomes has not yet been conducted. This study addresses this gap, aiming to provide critical insights into their comparative efficacy.

The present retrospective study aims to evaluate the long-term effects of SGLT2 inhibitors versus allopurinol on sUA levels in patients with T2DM over a 36-month follow-up, with particular attention to changes in estimated glomerular filtration rate (eGFR) and major clinical outcomes. By comparing these therapeutic approaches, this study seeks to elucidate whether SGLT2 inhibitors offer a multifaceted advantage in managing T2DM and CKD, potentially reducing therapeutic inertia and easing polypharmacy by addressing multiple risk factors with a single agent.

## 2. Materials and Methods

### 2.1. Study Design and Patient Selection

This retrospective cohort study was conducted using data collected in an outpatient diabetes clinic at a secondary healthcare hospital in Hungary (Hódmezővásárhely-Makó Healthcare Center, Makó). Caucasian male and female patients aged 18 years or older who were diagnosed with T2DM and in whom treatment with SGLT2 inhibitors was initiated between 1 January 2017 and 1 January 2020 were eligible. In Hungary, only empagliflozin (Jardiance^®^; Boehringer Ingelheim International, Ingelheim am Rhein, Germany) and dapagliflozin (Forxiga^®^; AstraZeneca AB, Södertälje, Sweden) are commercially available; therefore, two separate treatment cohorts were established, including 70 patients treated with 10 mg/day empagliflozin and 78 patients treated with 10 mg/day dapagliflozin.

An active comparator group was also established, consisting of 66 patients with T2DM who were initiated on allopurinol 100 mg/day (standard starting dose) for asymptomatic HUA during the study period; this number represented the patients who met the inclusion criteria. HUA was defined according to international recommendations as a sUA level > 416 µmol/L (7 mg/dL) in men and >357 µmol/L (6 mg/dL) in women [27].

Patients were excluded if their baseline sUA was below the sex-specific cut-off values, if they received concomitant treatment with allopurinol and an SGLT2 inhibitor, if empagliflozin was escalated from 10 to 25 mg/day or allopurinol from 100 mg/day to higher doses during follow-up, or if clinical data were incomplete.

The first outpatient visit at which treatment with an SGLT2 inhibitor or allopurinol was initiated was defined as the baseline visit (BV), and these measurements were taken as baseline values. Data were collected from the follow-up visits conducted 3, 6, 12, 24, and 36 months after the BV. Figure 1a presents the study flowchart, together with the exclusion criteria and the respective case numbers.

The study was conducted in accordance with the revised 2008 Helsinki Declaration. The study protocol was approved by the Institutional Review Board of Hódmezővásárhely-Makó Healthcare Center and the Hungarian National Public Health Center Institutional Committee of Science and Research Ethics (NNGYK/65897-2/2025, Approval date: 28 October 2025). All data were fully anonymized. Patient consent was waived due to the retrospective nature of data collection, in which no personal data were disclosed.

### 2.2. Data Collection

In all patients, the following data were collected in BV: age; sex; height; body weight; T2DM duration; antidiabetic therapy; fasting plasma glucose, total cholesterol; triglyceride; high-density lipoprotein; low-density lipoprotein cholesterol; left ventricular ejection fraction; and blood pressure. Serum uric acid and eGFR were collected at all visits. CKD stage was determined according to KDIGO recommendations [28], based on eGFR. The time dependence was determined by calculating the changes in parameters between the BV and the 36-month visit.

The medical history and the diagnoses of hypertension, chronic coronary artery disease, active malignant disease, diabetic neuropathy, and diabetic retinopathy were recorded. During follow-up visits, discontinuation of therapy, hospitalizations, new CV and kidney events, and mortality were recorded. Study design and protocol can be seen on Figure 1b.

### 2.3. Study Endpoints

The primary endpoint was the change in sUA from baseline to 36 months in each treatment arm, considered as a surrogate biomarker of treatment efficacy. Co-primary endpoints included changes in eGFR over 36 months and clinical outcomes such as treatment discontinuation, hospitalizations, and new CV and kidney events. Safety outcomes, including adverse events and all-cause mortality, were also assessed.

### 2.4. Statistical Analysis

Analyses were performed using GraphPad Prism (version 8.0.1; GraphPad Software, Boston, MA, USA) and SigmaStat (version 13; Systat Software, San Jose, CA, USA). Normality was assessed with the Shapiro–Wilk test. Normally distributed variables are presented as mean ± SD, while skewed non-normal variables are displayed as median (25th–75th percentile). Categorical variables are summarized as n (%), with between-group comparisons by Fisher’s exact test. Between-group comparisons at BV and at single follow-up time points were performed using one-way analysis of variance (ANOVA) with Tukey’s *post hoc* test for normally distributed variables, or the Kruskal–Wallis test with Dunn’s *post hoc* test for non-normal variables. Longitudinal analyses of continuous outcomes were conducted using a mixed-effects model fitted with residual maximum likelihood, with fixed effects for treatment group, time, and their interaction, and subject as the repeated factor. This approach was chosen to retain information from patients who discontinued therapy or deceased; events that are clinically relevant and not missing at random. For prespecified baseline-anchored contrasts, Dunnett’s adjustment was applied; for all-pairwise contrasts, either the Holm–Sidak or Tukey adjustment was applied, depending on whether the data met the assumptions required for ANOVA. Therapy adherence and mortality were analyzed as time-to-event outcomes using Kaplan–Meier survival analysis. Group differences in survival curves were evaluated with the log-rank (Mantel–Cox) test as the primary comparison, supplemented by the Gehan–Breslow–Wilcoxon test to account for early events. A log-rank test for trend was also performed across the three treatment arms. In addition, pairwise log-rank comparisons with Holm–Sidak correction were conducted to identify differences between individual treatment groups. Effect sizes were expressed as hazard ratios (Mantel–Haenszel and log-rank estimates) with 95% confidence intervals. Correlation analyses were performed using the Spearman method. To minimize statistical bias, the analysis of Δ values included only those patients who were followed until the end of the 36-month study period. A *p* value of <0.05 was considered to indicate statistical significance.

## 3. Results

### 3.1. Baseline Clinicopathologic Characteristics

The baseline clinical and pathologic characteristics (Table 1) were not significantly different between the patients treated with empagliflozin and dapagliflozin. However, HbA1c levels were significantly lower in the allopurinol group than in those treated with dapagliflozin (*p* < 0.001), reflecting the difference in the indications for initiation of the specific therapy.

### 3.2. Longitudinal Changes in sUA

Mixed-effects analysis revealed a significant main effect of time (*p* < 0.001) and treatment group (*p* = 0.006), indicating that sUA levels changed significantly over the 36-month follow-up and differed across treatment arms (Figure 2a). However, the time × treatment interaction was not significant, suggesting that the temporal trajectories of sUA were similar among all groups. At BV, sUA levels did not differ significantly between the three groups. At the 3-month visit, allopurinol treatment was associated with significantly lower sUA compared with both dapagliflozin (*p* = 0.021) and empagliflozin (*p* < 0.001). The difference between allopurinol and empagliflozin remained significant at 6 months (*p* < 0.001), 12 months (*p* = 0.009), and 24 months (*p* = 0.043), but not at 36 months. No significant differences were observed between empagliflozin and dapagliflozin at any time point.

### 3.3. Longitudinal Changes in eGFR

Both time (*p* < 0.001) and treatment group (*p* = 0.024) had significant effects on eGFR, with a significant time × treatment interaction (*p* < 0.001), indicating that the trajectories of eGFR differed between the treatment arms (Figure 2b). During follow-up, empagliflozin-treated patients consistently preserved higher eGFR compared with the allopurinol group, reaching statistical significance at 12 months (*p* = 0.044), 24 months (*p* < 0.001), and 36 months (*p* = 0.002). No significant differences were detected between dapagliflozin and the other groups at any time point.

In the allopurinol arm, eGFR declined progressively, with a significant reduction at the 3-month visit (−1 mL/min/1.73 m^2^, *p* = 0.023) and a cumulative loss approaching 10 mL/min/1.73 m^2^ at the 36-month visit (*p* < 0.001). Empagliflozin showed a transient decline at the 6-month visit (−4 mL/min/1.73 m^2^, *p* = 0.013), after which eGFR remained stable. Dapagliflozin demonstrated a modest early decrease in eGFR at the 3-month visit (*p* < 0.001), but no further significant changes thereafter.

The changes in the eGFR based CKD stages, mortality, and therapy discontinuation during the study period are illustrated by alluvial plots in Appendix A.

### 3.4. Cumulative Mortality

Kaplan–Meier analysis of all-cause mortality (Figure 3a) revealed significant differences between treatment groups (*p* = 0.042). Event rates were 13/66 (20%) in the allopurinol group, 5/78 (6%) in the dapagliflozin group, and 5/70 (7%) in the empagliflozin group. Both the log-rank test for trend (*p* = 0.038) and the Gehan–Breslow–Wilcoxon test (*p* = 0.046) confirmed worse survival in the allopurinol arm, whereas median survival was not reached in any group during the 36-month follow-up.

Pairwise comparisons showed that dapagliflozin was associated with significantly better survival compared with allopurinol (HR 2.9, 95% CI 1.12–7.57, *p* = 0.034), corresponding to an almost threefold higher mortality risk in the allopurinol group. The comparison between empagliflozin and allopurinol indicated a similar trend (HR 2.7, 95% CI 1.03–6.92), which showed a trend toward statistical significance (*p* = 0.052). No survival difference was observed between dapagliflozin and empagliflozin (HR 0.92, 95% CI 0.26–3.18, *p* = 0.889).

### 3.5. Therapy Adherence

Kaplan–Meier analysis of therapy discontinuation (Figure 3b) revealed significant differences among treatment arms (*p* = 0.004). Event rates were lowest in the allopurinol group (7/66, 10.6%), while discontinuations were markedly higher with dapagliflozin (28/78, 36.3%) and empagliflozin (20/70, 28.5%). Both the log-rank trend test (*p* = 0.014) and the Gehan–Breslow–Wilcoxon test (*p* = 0.004) confirmed reduced long-term adherence in the SGLT2 inhibitor groups.

Pairwise comparisons demonstrated significantly higher discontinuation risk with both dapagliflozin (HR 3.7, 95% CI 1.9–7.2, *p* < 0.001) and empagliflozin (HR 3.0, 95% CI 1.4–6.4, *p* = 0.007) compared with allopurinol, whereas no significant difference was observed between the two SGLT2 inhibitor arms (HR 1.2, 95% CI 0.67–2.11, *p* = 0.54).

### 3.6. Adverse Events, Detailed Mortality, Hospitalization, and Treatment Adherence

The proportions of adverse events, hospitalizations, and patients who discontinued therapy were not significantly different between the empagliflozin and dapagliflozin groups (Table 2). The incidence of acute coronary syndrome differed significantly between groups (*p* = 0.032). CV and kidney mortality was also higher in the allopurinol group compared with SGLT2 inhibitors (*p* = 0.026), mainly due to chronic heart failure (*p* < 0.001). Hospitalization occurred more frequently in patients treated with allopurinol (*p* = 0.045), while the time to first hospitalization showed no significant difference. Therapy discontinuation significantly differed across groups (*p* = 0.001), with urogenital infections (*p* = 0.001) and financial reasons (*p* = 0.003) contributing in the SGLT2 inhibitor groups, whereas dissatisfaction with therapy was observed only with allopurinol (*p* < 0.001).

## 4. Discussion

In this first study directly comparing two SGLT2 inhibitors with allopurinol regarding their impact on sUA and eGFR over a 36-month follow-up period, our analyses confirm the significant urate-lowering effect of SGLT2 inhibitors and highlight their potential benefits in the management of HUA in patients with T2DM. During patient enrollment, the baseline parameters were highly homogeneous across all three study groups. However, the significantly lower baseline HbA1c levels observed in the allopurinol group compared with the dapagliflozin group is clinically relevant, as therapy escalation with allopurinol was implemented primarily for the management of HUA and did not require intervention due to T2DM.

Patients with CKD, T2DM, and CV comorbidities are disproportionately affected by polypharmacy, which is associated with reduced adherence, increased drug–drug interactions, and a higher risk of adverse events [1]. Importantly, accumulating evidence suggests that polypharmacy itself may contribute to accelerated disease progression. In this fragile population, therapeutic strategies that provide pleiotropic benefits while minimizing additional pharmacological burden display particular clinical value [2]. SGLT2 inhibitors might represent an opportunity for de-prescribing, as these agents confer robust renal and CV protection and induce a consistent reduction in sUA levels. This approach offers a pragmatic means of mitigating polypharmacy while achieving meaningful urate reduction and renal protection.

In patients with an established indication for SGLT2 inhibition—such as CKD, T2DM, or chronic heart failure—these agents may reasonably be considered within a broader, individualized management strategy that also addresses hyperuricemia. However, in cases where SGLT2 inhibitor therapy alone is insufficient to achieve target serum uric acid levels, or in patients without an indication for SGLT2 inhibition in whom urate-lowering therapy with allopurinol is initiated but remains inadequate, combination therapy may be considered on an individualized basis. A stepwise approach to treatment intensification may help balance urate control with the potential systemic benefits associated with SGLT2 inhibition.

While epidemiological data link HUA to both CV and kidney disease, it is largely considered an epiphenomenon in CV pathology, whereas experimental and clinical evidence suggest a direct nephrotoxic effect by tubulointerstitial damage contributing to CKD progression [29,30]. However, the international guidelines for the treatment of CKD recommend urate-lowering therapy only in patients with symptomatic HUA [31]. Allopurinol is the most widely used urate-lowering agent; nevertheless, despite clear guideline recommendations against its use in asymptomatic HUA, it continues to be commonly prescribed in this setting. However, its inappropriate use raises several concerns, including potential adverse effects, decreased therapeutic efficacy, unnecessary financial burden, and suboptimal dosing due to conservative strategies for initial dosing [14,32]. In contrast, SGLT2 inhibitors have demonstrated broad CV and kidney protection in landmark trials [33,34,35,36,37,38]. SGLT2 inhibitors, such as empagliflozin and dapagliflozin, the two commercially available agents in Hungary, exert their therapeutic effects by inhibiting glucose reabsorption in proximal renal tubules [39]. Through improved urinary glucose excretion, SGLT2 inhibitors indirectly reduce sUA levels by competing with urate and glucose for reabsorption via the renal transporters [40,41]. Osmotic diuresis likely contributes to uricosuria; SGLT2 inhibitors also exert hemodynamic and anti-inflammatory effects that support kidney outcomes, though their impact on urate metabolism is indirect [42]. The distinct biological effects of allopurinol and SGLT2 inhibitors are clear implications of the chemical structural, pharmacodynamic, and pharmacokinetic properties of these agents (see details in Table 3).

Analysis of sUA level changes revealed broadly parallel trends across the three treatment arms. Allopurinol produced lower values than both empagliflozin and dapagliflozin at several interim assessments, yet these differences diminished over time. SGLT2 inhibitor-associated organ protection, reflected by preservation of eGFR, particularly with empagliflozin, was unequivocally evident in the present study. A transient decrease in eGFR observed in both the empagliflozin and dapagliflozin groups was an expected consequence of the reduction in glomerular hyperfiltration [43]. However, the eGFR subsequently stabilized, after which it remained stable. In contrast, a clear decline in eGFR was observed in the allopurinol group during the follow-up period. Although the sUA levels significantly decreased during the study, it was not sufficient to ensure long-term organ protection, as reflected in our findings.

Our findings regarding hospitalization and mortality rates were consistent with a more favorable CV and renal profile with SGLT2 inhibitors. While external evidence implicates multiple mechanisms—including modest natriuretic/osmotic diuresis and hemodynamic effects—in the benefits of SGLT2 inhibitors, our study did not evaluate acute decongestion, intravascular volume changes, or length of stay; therefore, causal interpretation should be made with caution [44,45]. Treatment discontinuation was least frequent with allopurinol; in the SGLT2 inhibitor arms, urogenital infections and financial reasons were commonly cited contributors to non-persistence.

The present study harbors several limitations that require further investigation. First, it was a single-center, retrospective cohort study conducted in a Caucasian population, which constrains generalizability and precludes causal inference due to potential confounding by indication and residual confounding. Importantly, allopurinol and SGLT2 inhibitors were prescribed for different primary indications, which may have influenced patient characteristics and clinical outcomes independently of treatment effects. Event counts for some endpoints were modest, limiting statistical power. Allopurinol was studied at a fixed starting dose of 100 mg/day without titration. Although no patient in the allopurinol group had symptomatic HUA or gout—so dose escalation would not have been guideline-concordant in this cohort—this dosing strategy limits inference about maximal urate-lowering potential in populations where titration would be appropriate. Outcomes such as adverse events, cause of death, and urogenital infections were ascertained retrospectively, introducing potential misclassification or under-reporting. Formal risk-adjusted analyses were not performed, and the study did not formally evaluate attainment of guideline-recommended sUA targets. Furthermore, substantial differences in treatment adherence led to markedly unequal group sizes over time; although we applied statistical methods designed to account for such imbalance, differential adherence may still represent an additional source of confounding. The absence of urinary albumin–to–creatinine ratio data limited the application of full KDIGO risk stratification (G–A staging) and may have obscured albuminuria-mediated treatment effects. Finally, eGFR and sUA were measured at scheduled visits only, and unmeasured factors (e.g., diuretic use, diet, intercurrent illness) could have affected laboratory values.

## 5. Conclusions

To our knowledge, this is the first study to directly compare allopurinol and SGLT2 inhibitors with respect to urate-lowering effects. In patients with T2DM and HUA, long-term trends in sUA reduction were comparable across all groups. Unlike allopurinol, SGLT2 inhibitors preserved kidney function, while no meaningful differences were observed between empagliflozin and dapagliflozin across the endpoints assessed. Mortality and hospitalizations occurred more frequently with allopurinol, whereas SGLT2 inhibitors were associated with more favorable clinical outcomes, highlighting their multidomain benefits beyond glucose control.

## Figures and Tables

**Figure 1 medsci-14-00012-f001:**
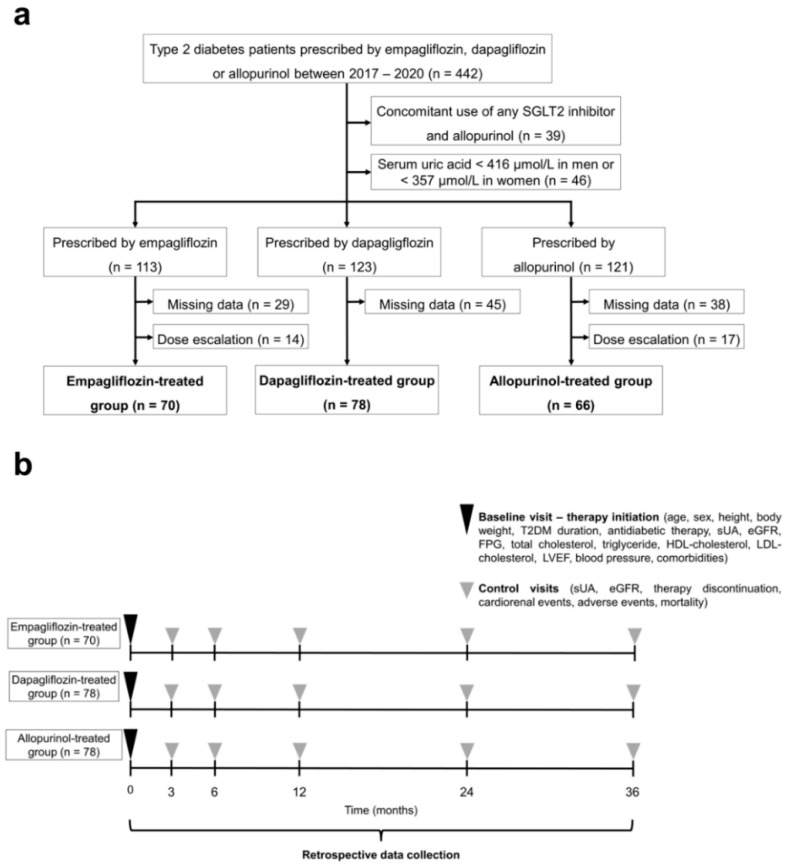
(**a**) Schematic flowchart showing the exclusion criteria used for patient selection. (**b**) Study design for retrospective data collection. Abbreviations: SGLT2, sodium-glucose cotransporter 2; T2DM, type 2 diabetes mellitus sUA, serum uric acid; eGFR, estimated glomerular filtration rate; FPG, fasting plasma glucose; HDL, high density lipoprotein; LDL, low density lipoprotein, LVEF, left ventricular ejection fraction.

**Figure 2 medsci-14-00012-f002:**
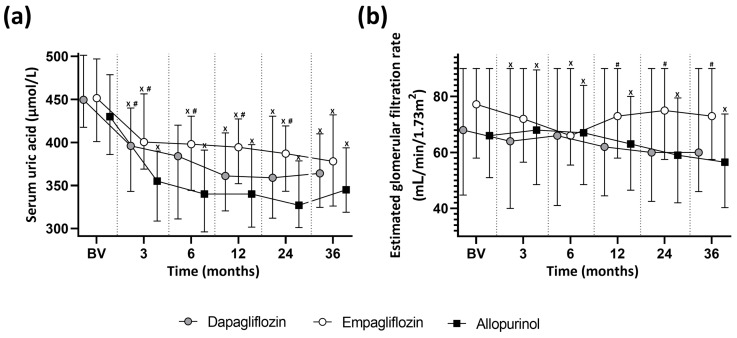
Longitudinal changes in serum uric acid levels (**a**) and in estimated glomerular filtration rate (**b**). Data are displayed as medians with 25th–75th percentiles. A gray full circle represents the dapagliflozin-treated group, the empagliflozin-treated group by a white full circle, and the allopurinol-treated group by a black full square ^x^ *p* < 0.05 vs. the baseline visit within the same treatment arm; ^#^ *p* < 0.05 vs. allopurinol within the same visit. Annotation symbols do not indicate the exact level of statistical significance; corresponding *p* values can be found in Appendix A.

**Figure 3 medsci-14-00012-f003:**
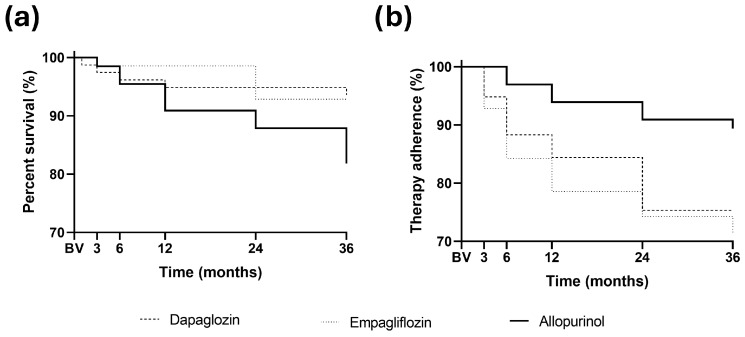
Kaplan–Meier analysis of all-cause mortality (**a**) and therapy adherence (**b**). Changes are expressed as percentages of the BV case numbers across all groups. The allopurinol-treated group is shown with a solid line, the dapagliflozin-treated group with a dashed line, and the empagliflozin-treated group with a dotted line.

**Table 1 medsci-14-00012-t001:** Clinicopathologic characteristics of patients enrolled in the study. Data are presented as medians with 25th–75th percentiles. *p* values of <0.05 were considered to indicate statistical significance. * *p* < 0.05 vs. dapagliflozin.

	Dapagliflozin (*n* = 78)	Empagliflozin (*n* = 70)	Allopurinol (*n* = 66)	*p* Value
Age (years)	63.0 (54.8; 68.3)	61.5 (54.0; 68.3)	64.0 (57.0; 68.3)	0.717
Diabetes duration (years)	10.0 (4.0; 14.3)	10.0 (6.0; 15.0)	10.5 (4.8; 18.3)	0.608
Body weight (kg)	101.0 (85.8; 115.0)	96.0 (83.0; 114.3)	104.0 (89.8; 113.5)	0.218
Body mass index (kg/m^2^)	30.5 (28.0; 35.5)	31.8 (27.5; 36.1)	32.7 (30.1; 36.8)	0.053
Fasting plasma glucose (mmol/L)	9.3 (7.2; 11.2)	9.3 (7.9; 12.2)	8.9 (8.5; 9.5)	0.722
HbA_1c_ (%)	8.5 (7.9; 9.0)	8.2 (7.6; 8.7)	7.8 (7.5; 8.1) *	* <0.001
Total cholesterol (mmol/L)	5.2 (4.4; 6.0)	5.5 (4.4; 6.1)	5.8 (4.6; 6.3)	0.188
HDL-cholesterol (mmol/L)	1.3 (1.1; 1.6)	1.3 (1.1; 1.7)	1.4 (1.1; 1.6)	0.495
LDL-cholesterol (mmol/L)	3.4 (2.6; 3.9)	3.4 (2.7; 4.0)	4.0 (2.8; 4.5)	0.061
Serum uric acid (μmol/L)	450 (418; 501)	452 (401; 497)	438 (388; 510)	0.813
Estimated glomerular filtration rate (mL/min/1.73 m^2^)	68.0 (45.0; 90.0)	77.2 (58.0; 90.0)	75.0 (55.0; 90.0)	0.300
Left ventricular ejection fraction (%)	56.0 (50.0; 60.0)	55.0 (45.0; 60.0)	60.0 (40.0; 60.0)	0.602
Insulin usage (*n*, %)				
No insulin usage	32 (41.0%)	29 (41.4%)	39 (59.1%)	0.063
Basal insulin supported oral therapy	7 (8.97%)	7 (10.0%)	3 (4.54%)	0.088
Fixed-ratio combination therapy	1 (1.28%)	4 (5.71%)	0 (0.00%)	0.079
Multiple-dose injection therapy	38 (48.7%)	30 (42.9%)	24 (36.4%)	0.333
Non-insulin antidiabetic drug usage (*n*, %)				
Metformin	59 (75.6%)	51 (72.9%)	49 (74.2%)	0.923
Sulfonylurea	19 (24.4%)	18 (25.7%)	20 (30.3%)	0.717
Dipeptidyl peptidase-4 inhibitor	11 (14.1%)	6 (8.57%)	14 (21.2%)	0.111
Glucagon-like peptide-1 receptor agonists	1 (1.28%)	1 (1.43%)	5 (7.56)%	0.097
Comorbidities (*n*, %)				
Hypertension	44 (56.4%)	47 (67.1%)	48 (72.7%)	0.113
Chronic coronary artery disease	21 (26.9%)	15 (21.4%)	20 (30.3%)	0.651
Diabetic retinopathy	14 (17.9%)	11 (15.7%)	15 (22.7%)	0.574
Peripheric sensory neuropathy	25 (32.1%)	32 (45.7%)	29 (43.9%)	0.310
Chronic kidney disease	31 (39.7%)	21 (30.0%)	33 (50.0%)	0.058
Active malignancy	2 (2.56%)	2 (2.86%)	1 (1.51%)	1.000

**Table 2 medsci-14-00012-t002:** Mortality, hospitalization, adverse events and adherence of patients enrolled in the study. Student’s *t* test and chi-square test were used to compare the dapagliflozin and empagliflozin treatment arms, as appropriate. Data are presented as medians with 25th–75th percentiles or as n with percentages. *p* values of <0.05 were considered to indicate statistical significance. * *p* < 0.05 vs. dapagliflozin.

		Dapagliflozin (*n* = 78)	Empagliflozin (*n* = 70)	Allopurinol (*n* = 66)	*p* Value
Acute coronary syndrome during follow-up (*n*, %)		2 (2.6%)	4 (5.7%)	9 (13.6%)	0.032
Mortality due to cardiovascular and kidney events (*n*, %)	All (*n*, %)	5 (6.4%)	5 (7.1%)	13 (19.7%)	0.026
	Acute coronary syndrome (*n*)	1	4	3	0.320
	Sudden cardiac arrest (*n*)	2	0	0	0.330
	Cerebrovascular disease (*n*)	1	1	1	1.000
	Chronic heart failure (*n*)	1	0	9	0.0001
Hospitalized patients (*n*, %)		10 (12.8%)	9 (12.9%)	18 (27.3%) *	* 0.035
Total hospitalization (incidence/3 years)		3.33	3.00	6.00	-
Total hospitalization (incidence/100 patients/year)		4.27	4.29	9.09	-
Time of first hospitalization (months)		12 (9; 18)	12 (3; 24)	6 (3; 12)	0.100
Cause of first hospitalization	Acute coronary syndrome (*n*)	2	2	5	0.299
	Chronic heart failure (*n*)	5	4	10	0.127
	Cerebrovascular disease (*n*)	1	2	2	0.739
	Urinary tract infection (*n*)	2	0	0	0.331
	Other cause (*n*)	0	1	1	0.534
Therapy discontinuation (*n*, %)		28 (35.9%)	20 (28.6%)	7 (10.6%)	0.001
Cause of therapy discontinuation	Worsening kidney function (*n*)	3	1	0	0.391
	Polyuria (*n*)	2	3	0	0.327
	Urogenital infections (*n*)	12	6	0	0.001
	Lower abdominal pain (*n*)	3	0	0	0.109
	Diarrhea (*n*)	1	0	0	0.999
	Financial causes (*n*)	7	10	0	0.003
	Therapeutic dissatisfaction (*n*)	0	0	7	<0.001

**Table 3 medsci-14-00012-t003:** Comparative structural, pharmacological, and urate-lowering characteristics of empagliflozin, dapagliflozin, and allopurinol.

Feature	Empagliflozin	Dapagliflozin	Allopurinol
Drug class	Sodium–glucose cotransporter 2 inhibitor	Sodium–glucose cotransporter 2 inhibitor	Xanthine oxidase inhibitor
Core chemical scaffold	C-aryl glucoside	C-aryl glucoside	Purine analog
Key structural motif	Glucose moiety linked via C–C bond to substituted aromatic ring	Glucose moiety linked via C–C bond to substituted aromatic ring	Pyrazolo[3,4 d]pyrimidine ring
Key structural distinction	Larger hydrophobic aromatic substitutions	Smaller, more polar aromatic substitutions	Smaller, more polar aromatic substitutions
SGLT2 receptor selectivity	~2500×	~1200×	n/a
Active metabolite	None clinically relevant	None clinically relevant	Oxypurinol (active, renally cleared)
Mechanism of uric acid lowering	Indirect: increased renal uric acid excretion via glycosuria-induced uricosuria	Indirect: increased renal uric acid excretion via glycosuria-induced uricosuria	Direct: inhibition of uric acid production
Renal handling relevance	Acts on proximal tubular glucose reabsorption; secondary effect on urate transporters	Acts on proximal tubular glucose reabsorption; secondary effect on urate transporters	Alters purine metabolism; renal excretion of urate and oxypurinol
Structure–activity relationship	Aromatic substitutions enhance SGLT2 selectivity and potency.	Aromatic substitutions enhance SGLT2 selectivity and potency.	Structural mimicry enables competitive enzyme inhibition
Direct urate receptor binding	No	No	No
Metabolic degradation	Minimal	Minimal	Hepatic + renal
Onset of urate-lowering effect	Gradual, dependent on metabolic and renal effects	Gradual, dependent on metabolic and renal effects	Direct and rapid enzyme-level effect

## Data Availability

Data are available from the corresponding authors upon reasonable request; however, access is restricted due to ethical approval requirements.

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
