# Peer review of "Comparison of the Efficacy of Empagliflozin, Dapagliflozin, and Allopurinol Based on Serum Uric Acid Levels and Kidney Function in Patients with Type 2 Diabetes Mellitus: A Retrospective Cohort Study"

_medsci, 2025, doi:10.3390/medsci14010012_

Round 1
Reviewer 1 Report
Comments and Suggestions for Authors
The stated purpose of this paper is to see if SGLTs lower uric acid to a similar level as allopurinol. The authors show - over the long run the reduction in levels is comparable. The authors further show the SGLTs provide renal protection whereas allopurinol does not. This is the paper short and sweet!!
The authors then go on to other subjects like mortality, cost of SGLTs, etc that are not related to the purpose of the paper. All these extra subjects can be deleted. Discussion of CVD mortality is suspect since numbers are small, the presence of CKD is higher in the allopurinol group compared to the SGLT groups, and no data is presented on baseline heart failure. No adjusted analyses are done
The paper should focus on its intended purpose. It can be short and to the point. You can discuss if SGLT can be used as a uricosuric agent (albeit expensive) if allopurinol is not tolerated in someone with DM. Would it decrease the pill count of such a person? Please note the first paragraph of the INTRO has no follow up and can be deleted unless you discuss this last point.
Finally, Fig 3A should be 3B and v.v.
Author Response
Comment: The stated purpose of this paper is to see if SGLTs lower uric acid to a similar level as allopurinol. The authors show - over the long run the reduction in levels is comparable. The authors further show the SGLTs provide renal protection whereas allopurinol does not. This is the paper short and sweet!
The authors then go on to other subjects like mortality, cost of SGLTs, etc that are not related to the purpose of the paper. All these extra subjects can be deleted. Discussion of CVD mortality is suspect since numbers are small, the presence of CKD is higher in the allopurinol group compared to the SGLT groups, and no data is presented on baseline heart failure.
Response: Thank you for your valuable and thoughtful comments. The primary endpoint of our study was the assessment of serum uric acid reduction; however, the analysis of clinical outcomes was prespecified as a secondary endpoint. In agreement with the concern regarding the inclusion of topics beyond the core scope of the manuscript (e.g., cost considerations), we have substantially shortened the Discussion section. Nevertheless, given the predefined secondary objectives, we briefly addressed mortality-related outcomes, albeit more concisely and cautiously in the revised version of the manuscript. We respectfully and sincerely thank you for your understanding.
Comment: No adjusted analyses are done
Response: Indeed, we have added it among the study limitations (ll. 388-390).
Comment: The paper should focus on its intended purpose. It can be short and to the point. You can discuss if SGLT can be used as a uricosuric agent (albeit expensive) if allopurinol is not tolerated in someone with DM. Would it decrease the pill count of such a person? Please note the first paragraph of the INTRO has no follow up and can be deleted unless you discuss this last point.
Response: A paragraph about polypharmacy is now added to the discussion session (ll. 288-306).
Comment: Finally, Fig 3A should be 3B and v.v.
Response: Done.
Reviewer 2 Report
Comments and Suggestions for Authors
An interesting study, which is apparently OK regarding sample size, although the latter is not justified in the text and the authors should see to it in a revision. Though not a head to head comparison the following study -- You, Y., Zhao, Y., Chen, M. et al. Effects of empagliflozin on serum uric acid level of patients with type 2 diabetes mellitus: a systematic review and meta‐analysis. Diabetol Metab Syndr 15, 202 (2023). https://doi.org/10.1186/s13098-023-01182-y -- would probably merit to be cited.
Author Response
Comment: An interesting study, which is apparently OK regarding sample size, although the latter is not justified in the text and the authors should see to it in a revision. Though not a head to head comparison the following study -- You, Y., Zhao, Y., Chen, M. et al. Effects of empagliflozin on serum uric acid level of patients with type 2 diabetes mellitus: a systematic review and meta‐analysis. Diabetol Metab Syndr 15, 202 (2023). https://doi.org/10.1186/s13098-023-01182-y -- would probably merit to be cited.
Response: Thank you for the positive comment. The recommended publication has been included in the revised manuscript as reference 41.
Reviewer 3 Report
Comments and Suggestions for Authors
- The abstract states that the long-term antihyperuricemic effects of SGLT2 inhibitors are lesser known, but primarily a urate-lowering therapy is an appropriate comparator in this retrospective setting.
- As this is a retrospective cohort, confounding by indication, adherence differences, or comorbidity burden may influence mortality and eGFR outcomes. A brief statement acknowledging these limitations would strengthen transparency.
- The abstract does not state whether empagliflozin/dapagliflozin and allopurinol were administered at standardized or variable doses, which may impact comparative effectiveness.
- The observation that allopurinol users had higher mortality/hospitalization rates could reflect underlying disease severity rather than treatment effect. The authors should clarify whether risk adjustment was applied.
- Add a structure similarity of all three drugs with structure and structure activity relationship with biological activity.
- Claims such as “SGLT2 inhibitor therapy offers multidomain benefits” should be toned down or qualified, given the observational, retrospective design that limits causal interpretation.
- Several sentences contain grammatical inconsistencies (e.g., “their long-term antihyperuricemic impact is lesser unknown”) or awkward phrasing. A language revision would improve clarity and professionalism.
- While numerical reductions are presented, it is unclear whether these changes achieved clinically meaningful thresholds or whether target uric acid levels were met across groups.
Author Response
1. comment: The abstract states that the long-term antihyperuricemic effects of SGLT2 inhibitors are lesser known, but primarily a urate-lowering therapy is an appropriate comparator in this retrospective setting.
Response: Thank you for your comment. We have revised the problematic statement in the abstract accordingly (ll. 19-20).
2. comment: As this is a retrospective cohort, confounding by indication, adherence differences, or comorbidity burden may influence mortality and eGFR outcomes. A brief statement acknowledging these limitations would strengthen transparency.
Response: Thank you. We have addressed the expected limitations arising from the different treatment indications and from the substantially different adherence observed during the study period (ll. 379-381). We believe, however, that at the time of patient selection, the comorbidity profiles were sufficiently comparable across the study groups, as demonstrated in Table 1. Naturally, disease progression during follow-up differed between groups (see clinical endpoints); however, this constitutes an inherent part of the study outcomes. Therefore, we respectfully request your understanding regarding our decision not to further elaborate on this aspect in the manuscript.
3. comment: The abstract does not state whether empagliflozin/dapagliflozin and allopurinol were administered at standardized or variable doses, which may impact comparative effectiveness.
Response: Done.
4. comment: The observation that allopurinol users had higher mortality/hospitalization rates could reflect underlying disease severity rather than treatment effect. The authors should clarify whether risk adjustment was applied.
Response: The higher mortality and hospitalization rates observed in the allopurinol group may, at least in part, reflect differences in underlying disease severity. Notably, SGLT2 inhibitors are well established to confer cardiovascular and cardiorenal protection, and a signal consistent with these benefits was also observed in our cohort. The direction and magnitude of the observed outcomes are in line with previously published data, supporting the plausibility that the differences may be driven, at least in part, by the effects of SGLT2 inhibition. This interpretation is further supported by the contemporaneous patient inclusion and the relatively homogeneous population, which may have reduced—but not eliminated—between-group heterogeneity.
Nevertheless, residual confounding cannot be excluded. Formal risk-adjusted analyses were not performed, which we acknowledge as an important limitation and have now explicitly stated in the revised manuscript. Information is now added (ll. 388-393).
5. comment: Add a structure similarity of all three drugs with structure and structure activity relationship with biological activity.
Response: The requested information is now added as a novel Table 3.
6. comment: Claims such as “SGLT2 inhibitor therapy offers multidomain benefits” should be toned down or qualified, given the observational, retrospective design that limits causal interpretation.
Response: Done.
7. comment: Several sentences contain grammatical inconsistencies (e.g., “their long-term antihyperuricemic impact is lesser unknown”) or awkward phrasing. A language revision would improve clarity and professionalism.
Response: A full linguistic review covering the entire material has been carried out.
8. comment: While numerical reductions are presented, it is unclear whether these changes achieved clinically meaningful thresholds or whether target uric acid levels were met across groups.
Response: The study did not formally evaluate attainment of guideline-recommended serum uric acid targets, limiting assessment of clinical meaningfulness beyond numerical reductions. Information is added (ll. 388-393).
Round 2
Reviewer 1 Report
Comments and Suggestions for Authors
The paper is improved but there are still some points that need clarification:
- On line 397ff you state none of the participants had gout / symptomatic HU. So why were they treated? You said at the onset, that KIDGO is against treatment of asymptomatic people. Also, you should mention in the METHODS that the allopurinol group was asymptomatic.
- Lines 319-390 mostly repeat your results. There is nothing new here. Please shorten or remove.
- Line 312 - "..their direct effect was indirect." Remove "direct."
Author Response
Comment: On line 397ff you state none of the participants had gout / symptomatic HU. So why were they treated? You said at the onset, that KIDGO is against treatment of asymptomatic people. Also, you should mention in the METHODS that the allopurinol group was asymptomatic.
Response: Hyperuricaemia is a substantially overtreated condition, and adherence to the recommendations of professional societies in this area appears to be suboptimal in routine clinical practice. We address this aspect in detail in both the Introduction and the Discussion (see polypharmacy). Consequently, our retrospective patient population is not free from this clinical inertia. However, as this is a real-world study, we believe that this does not undermine the validity or interpretability of our results. The requested clarification has, of course, been incorporated into the Methods section.
Comment: Lines 319-390 mostly repeat your results. There is nothing new here. Please shorten or remove.
Response: The indicated part of the draft has been shortened.
Comment: Line 312 - "..their direct effect was indirect." Remove "direct."
Response: We apologize for the error; the word “direct” has been removed.
